# Introducing Beyond FA: Crowdsourcing Imaging Biomarkers for Alzheimer's Disease

**Elyssa M. McMaster**[1]                    ELYSSA.M.MCMASTER@VANDERBILT.EDU
[1] *Vanderbilt University, Nashville, TN, USA*
**Nancy R. Newlin**[1]                       NANCY.R.NEWLIN@VANDERBILT.EDU
**Trent Schwartz**[1]                        TRENT.SCHWARTZ.1@VANDERBILT.EDU
**Adam M. Saunders**[1]                      ADAM.M.SAUNDERS@VANDERBILT.EDU
**Gaurav Rudravaram**[1]                     GAURAV.RUDRAVARAM@VANDERBILT.EDU
**Yehyun Suh**[1]                            YEHYUN.SUH@VANDERBILT.EDU
**Jongyeon Yoon**[1]                         JONGYEON.YOON@VANDERBILT.EDU
**Michael E. Kim**[1]                        MICHAEL.KIM@VANDERBILT.EDU
**Chloe Cho**[1]                             CHLOE.CHO@VANDERBILT.EDU
**Karthik Ramadass**[1]                      KARTHIK.RAMADASS@VANDERBILT.EDU
**Yihao Liu**[1]                             YIHAO.LIU@VANDERBILT.EDU
**Lianrui Zuo**[1]                           LIANRUI.ZUO@VANDERBILT.EDU
**Eleftherios Garyfallidis**[4]              ELEF@INDIANA.EDU
[4] *Indiana University, Bloomington, IN, USA*
**Talia M. Nir**[3]                          TNIR@USC.EDU
[3] *Mark and Mary Stevens Neuroimaging and Informatics Institute, Keck School of Medicine of the University of Southern California*
**Neda Jahanshad**[3]                        NJAHANSH@USC.EDU
**Kurt G. Schilling**[2]                     KURT.G.SCHILLING.1@VUMC.ORG
[2] *Vanderbilt University Medical Center, Nashville, TN, USA*
**Daniel Moyer**[1]                          DANIEL.MOYER@VANDERBILT.EDU
**Bennett A. Landman**[1,2]                  BENNETT.LANDMAN@VANDERBILT.EDU

**Editors:** Accepted for publication at MIDL 2025

## Abstract

The inclusion of diffusion weighted magnetic resonance imaging (DW-MRI) in neuroimaging studies has enabled deeper understanding of white matter structure of the human brain. Fractional anisotropy (FA), a commonly used metric for assessing white matter integrity, offers high sensitivity but often suffers from low specificity in detecting pathology. FA's limitations as a biomarker are often reflected in its inconsistency across different age groups and disease stages, especially in heterogeneous populations. With the emergence of alternative white matter models and metrics—such as complex network measures and tract-specific tractography—researchers now have access to various methods for assessing white matter integrity, especially in the context of lower-quality and heterogeneous data. Given the growing number of white matter models and metrics, there is a pressing need to explore and evaluate these metrics in order to move beyond the constraints of FA. To address this, we introduce Beyond FA, an open challenge that invites teams to submit the imaging biomarker(s) of their choice. The goal is to compare and gain deeper understanding of white matter metrics and their association with Alzheimer's Disease (AD).

**Keywords:** Diffusion MRI, harmonization, microstructure, tractography, connectome, feature extraction

## 1. Introduction

Many international-scale research studies employ diffusion-weighted MRI (DW-MRI) as part of their imaging protocol, which invites an opportunity to study white matter (LaMontagne et al., 2019), (Glasser et al., 2013), (Ellis et al., 2009), (Archer et al., 2023), (Jack et al., 2008). However, due to the heterogeneous nature of both large-scale study scanner hardware and pathological biology, white matter models often suffer from high variability (Borrelli et al., 2023), (Erus et al., 2018), (Newlin et al., 2024). Fractional anisotropy (FA) is a diffusion tensor imaging (DTI) metric often considered in Alzheimer's Disease (AD) characterization, but its low specificity in pathological contexts introduces demand to consider alternatives (Farquharson et al., 2013), (Figley et al., 2022), (McMaster et al., 2024).

We hypothesize that white matter models beyond FA have the potential to classify age, sex, and cognitive status with higher accuracy when compared to FA with the same framework. We propose a challenge as a setting to collect and compare white matter models.

For our Beyond FA challenge, participants build a Docker that performs the white matter model computations of their choice and then extracts features from each image to correlate with age, sex, and cognitive status. As a separate evaluation, we will compute the correlation for other, secret variables that we will reveal at the conclusion of the challenge. The challenge is hosted on grand-challenge.org.

## 2. Data

In this challenge, participants may use any single shell diffusion data to develop their own method to extract image features that correlate with age, sex, and cognitive status, as shown in Figure 1. We will not reveal our evaluation data to participants to encourage robust approaches. Our data will be heterogeneous across sites and scanners. A comprehensive breakdown of the data used for evaluation is included in Figure 1, with our evaluation approach discussed in Section 3.

## 3. Submission Evaluation

Participants submit a Docker container with their algorithm for evaluation. Assessment will be performed with a shallow multi-layer perceptron (MLP). We evaluate based on age, sex, cognitive status, and two hidden variables revealed at the end of the challenge to avoid calibration to these variables. We train and test our MLP based on the feature vectors from each individual algorithm submission and the associated age, sex, cognitive status, and hidden variable labels. A successful challenge algorithm will extract meaningful features that correlate with our labels.

## 4. Broader Contributions

Here, we aim to address the limitations of data availability and scanner hardware variability to maximize the potential of large-scale white matter model findings related to AD. This challenge provides insight to techniques that are specific to AD, but our workflow may be applied to other pathology with multi-site or multi-acquisition data.

Table 1: The age, sex, and cognitive status breakdown for our training and testing data. The data imbalance between CN, MCI, and AD should challenge teams to thoughtfully design their algorithms with the paucity of AD data available compared to CN across major studies in mind.

|  | Training | Testing |
|---|---|---|
| **CN/MCI/AD** | 210/210/80 | 34/33/33 |
| **Male/Female** | 252/248 | 50/50 |
| **Age** (years) | $73.3 \pm 8.4$ | $72.7 \pm 7.3$ |

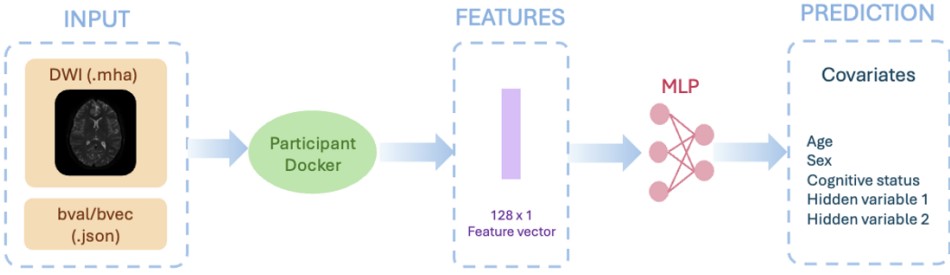

Figure 1: The overall architecture of our testing framework. Participants are only responsible for the Docker submission; everything else is provided by us on the grand-challenge.org page.

## Acknowledgments

The Vanderbilt Institute for Clinical and Translational Research (VICTR) is funded by the National Center for Advancing Translational Sciences (NCATS) Clinical Translational Science Award (CTSA) Program, Award Number 5UL1TR002243-03. This work was conducted in part using the resources of the Advanced Computing Center for Research and Education at Vanderbilt University, Nashville, TN. We would like to acknowledge support from U24AG074855, 1R01EB017230 , R01EB02758, R01MH134004, R01AG087513, AARG-23-1149996 and P50HD103537.

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
