# OpenReview forum: "Introducing Beyond FA: Crowdsourcing Imaging Biomarkers for Alzheimer’s Disease"
_MIDL.io/2025/Short_Papers — MIDL 2025 - Short Papers_

### Official Review · Reviewer_97CY · 2025-04-23

**Rating:** 5
**Confidence:** 5

**Summary:**

The authors present a challenge for extracting imaging biomarkers from DWI brain images. The challenge is hosted on grand-challenge.org, and it offers a large dataset with thorough submission evaluations. The task aligns well with the interests of MIDL.

**Strengths:**

The paper is well written and the task is clear, with instructions on how to get started.

**Weaknesses:**

This is just a hunch, but regarding the two hidden variables, in the abstract the authors mention FA's limitation "across different age groups and -----". I wonder if that second part hints at what those variables are. If so, perhaps they would like to remove that part.

The timeline of the challenge is unclear, the website only says that it closes in July 2025, depending on the exact date it might be more or less suitable for MIDL 2025.

---

### Decision · Program_Chairs · 2025-05-01

Accept